# Serum Androgen Metabolites Correlate with Clinical Variables in African and European American Men with Localized, Therapy Naïve Prostate Cancer

**DOI:** 10.3390/metabo13020284

**Published:** 2023-02-16

**Authors:** Swathi Ramakrishnan, Rick A. Kittles, Wendy J. Huss, Jianmin Wang, Kristopher Attwood, Anna Woloszynska

**Affiliations:** 1Department of Pharmacology and Therapeutics, Roswell Park Comprehensive Cancer Center, Buffalo, NY 14263, USA; 2Community Health and Preventive Medicine, Morehouse School of Medicine, Atlanta, GA 30310, USA; 3Department of Bioinformatics and BioStatistics, Roswell Park Comprehensive Cancer Center, Buffalo, NY 14263, USA

**Keywords:** prostate cancer, African American, European American, androgen metabolism, androgen receptor, health disparities

## Abstract

Dihydrotestosterone (DHT) and testosterone (T), which mediate androgen receptor nuclear translocation and target gene transcription, are crucial androgens and essential molecular triggers required for the proliferation and survival of prostate cancer cells. Therefore, androgen metabolism is commonly targeted in the treatment of prostate cancer. Using a high-pressure liquid chromatographic assay with tandem mass spectral detection, we determined the serum levels of metabolites produced during DHT/T biosynthesis in African American (AA) and European American (EA) men with localized, therapy naïve prostate cancer. Serum progesterone and related metabolites were significantly lower in AA men than in EA men, and these differences were associated with rapid disease progression. Multivariate analysis revealed significant differences between a subset of intermediate androgen metabolites between AA and EA men and between men with <=3 + 4 and >=4 + 3 Gleason score disease. AA men have a significantly higher frequency of single nucleotide polymorphisms in CYP11B1 and CYP11B2, enzymes that regulate corticosterone-aldosterone conversion. Finally, higher levels of T and pregnenolone were associated with a lower risk of progression-free survival only in AA men. This work provides new insight into androgen metabolism and racial disparities in prostate cancer by presenting evidence of dysregulated androgen biosynthesis in therapy naïve disease that correlates with clinical variables.

## 1. Introduction

Androgens, including DHT and T, are synthesized through an enzyme-mediated process that involves the conversion of several intermediate metabolites [1,2]. Subsequently, DHT and T bind to and initiate androgen receptor (AR) nuclear translocation, which results in AR target gene transcription that modulates cell cycle, apoptosis, cell differentiation, and other processes necessary for the development and progression of prostate cancer [3,4,5]. In addition, androgen-driven AR signaling and associated transcriptional activity have been found to be a focal signaling axis in prostate cancer [2,5,6].

Multiple cell types, including the Leydig cells of the testis and adrenal glands, produce DHT and T. DHT and T de novo synthesis requires cholesterol as the primary precursor [7]. Cholesterol is first converted to pregnenolone which is converted into several intermediate metabolites, including progesterone, dehydroepiandrosterone (DHEA), androstenedione (ASD), and androsterone (AND) [1,2]. DHT synthesized through DHEA to T conversion can be referred to as the primary pathway, while other mechanisms of androgen biosynthesis belong to multiple alternate pathways [1,2,8]. Alternate pathways of androgen biosynthesis are thought to occur either in response to castration or androgen deprivation therapy, but not in localized or therapy naïve disease prior to radical prostatectomy [1,9,10].

There are few studies that report race-specific differences in the levels of androgens and intermediate metabolites. In the non-diseased state, AA men have increased DHT levels in the serum compared with EA men [11,12,13]. High serum levels of total cholesterol are associated with a high risk of developing prostate cancer and are a significant risk factor for recurrent disease in AA men [14,15]. Compared with EA men, prostate cancer tissues from AA men have similar DHT and T levels but have significantly higher levels of intermediate androgen metabolites such as DHEA and ASD [16,17]. In castration recurrent disease, DHEA and T levels, among other intermediate metabolites, are higher compared to levels prior to castration in the same men [18]. Other studies show altered lipid metabolism, including triglycerides and phosphatidic acid levels, between AA and EA men with prostate cancer [19,20]. However, the race-specific differences of DHT, T, and intermediate androgen metabolites, specifically in the serum of men with localized, therapy naïve disease, are not clearly elucidated.

Despite notable progress in understanding the biological pathways that differ between AA and EA men with prostate cancer [21,22,23,24], a significant lack of key information exists concerning race-specific differences in androgen metabolism. Specifically, the relationship between androgen metabolites, disease characteristics, and survival outcomes in localized, therapy-naïve disease is lacking. Here, we utilized serum samples from localized, therapy naïve prostate cancer patients. We determined that significant differences in androgen metabolites exist between AA and EA men and investigated their relationship with clinical and pathological disease parameters. Our results showed a significant association of androgens and intermediate metabolites with progression-free survival and Gleason score in AA and EA men with localized, therapy-naïve prostate cancer.

## 2. Materials and Methods

All clinical samples were collected with informed consent before treatment/radical prostatectomy and at the time of diagnosis at Roswell Park. All samples were de-identified before analyses, and the demographics are listed in Appendix A. Ancestry informative markers (AIMs) [25] were performed to determine the concordance between self-reported race and AIMs (Appendix A). Most clinical samples showed a direct correlation between self-reported race and AIMs (Appendix A). 

### 2.1. Liquid Chromatographic Assay with Tandem Mass Spectral Detection (LC-MS/MS)

Samples were analyzed for androgens using high-pressure LC-MS/MS [26]. The time of collection does not affect the serum measurements of androgens [27]. Immediately after androgen analysis, extracted samples were re-injected using a semi-quantitative targeted metabolomics approach to examine 26 additional compounds within the steroidogenesis pathway. Serum samples were quantified using human serum calibration and quality control (QC) samples, prepared by spiking known amounts of androgens into charcoal-stripped female human serum (Bioreclamation) (Appendix A). Liquid/liquid sample extractions were performed by mixing 250 μL of a serum calibrator, QC, matrix blank, or study sample with LC/MS-grade water, internal standard (IS) solution (75.0 per 225 pg/mL d_3_-T/d_3_-DHT in 75% methanol), and 4.0 mL of methyl-tert-butyl ether (MTBE, EMD Millipore, Billerica, MA, USA). Tubes were capped with Teflon-lined caps, vortexed, and rotated for 15 min at room temperature; the tubes were then centrifuged (Heraeus Multifuge X3R, Thermo Scientific, Waltham, MA, USA, ) at 2800 rpm at 4 °C for 15 min to separate the liquid phases. The aqueous phase was frozen in a dry ice/acetone bath, and the MTBE layer was poured into a clean conical tube. The residue left after MTBE evaporation at 37 °C under nitrogen was reconstituted with 60 μL of 60% methanol. The resulting suspensions were centrifuged at 2800 rpm at 4 °C for 5 min to separate insoluble materials. The clear supernatant was transferred to an autosampler vial, and 7–15 μL aliquots were injected.

LC-MS/MS analysis of extracted samples was performed by using a Prominence UFLC System (Shimadzu Scientific Instruments, Kyoto, Japan) and QTRAP^®^ 5500 mass spectrometer (AB Sciex, Framingham, MA, USA), with an electrospray ionization source, and two 10-port switching valves (Valco Instruments Co., Inc., Houston, TX, USA, model EPC10W). Chromatographic separation was performed using a Phenomenex Luna C18(2) column (part number 00F-4251-B0) preceded by a Phenomenex SecurityGuard cartridge (C18, part number AJ0-4286), which was maintained at 60 °C and flow rate of 175 µL/min. Chromatography was performed by using a biphasic gradient (Mobile Phase A: 65% methanol containing 400 µL of 1.00 M ammonium formate and 62 µL/L of concentrated formic acid; and Mobile Phase B: 100% methanol with 400 µL of 1.00 M ammonium formate and 62 µL/L of concentrated formic acid). After standard androgen analysis, each sample was re-injected under a different set of gradient conditions to perform a semi-quantitative analysis of the steroidogenesis pathway. Analytes were detected with multiple reaction monitoring (MRM) in positive ion mode controlled by AB Sciex Analyst^®^ software, version 1.6.2. Mass spectrometer conditions for androgens were an ion spray voltage of 5250 volts, turbo gas temperature of 700 °C, gas 1 = 65, gas 2 = 60, curtain gas 20, and collision-associated dissociation (CAD) gas = medium. Q1 and Q3 were set at the unit mass resolution, and nitrogen was used. Direct infusion and flow injection analysis were used to optimize the voltages for maximum parent/fragment ion pair intensities. Minor modifications in operating conditions were performed to maintain optimal sensitivity.

Analyte/IS peak area response ratios versus nominal concentrations (ng/mL) and weighted linear regressions with a weighting factor 1/concentration were used to prepare calibration curves^2^. d_3_-T was used as the IS for T, ASD, and DHEA, and d_3_-DHT for DHT and AND. Back-calculated concentrations were generated using the formula x = (y−b)/m, where x represents the back-calculated concentration, y represents the analyte/IS peak area ratio, b represents the y-intercept, and m represents the slope. Calibrator and quality control acceptance criteria followed the U.S. Food and Drug Administration bioanalytical guidelines whereby all acceptable concentrations must have accuracy deviations of ≤±15% from the nominal concentration with relative standard deviations (% RSD) ≤15%, except at the lower limit of quantitation (LLOQ) where ≤20% deviations were allowed for both parameters. Performance data from the analysis of 855 human serum samples are provided in Appendix A.

### 2.2. Statistical Analyses

#### 2.2.1. Patient Demographics and Comparisons of Clinical Variables between AA and EA Men

Patient demographic and clinical characteristics, described in Appendix A, are reported by race using the mean, median, standard deviation, and range for continuous variables, and by using frequencies and relative frequencies for categorical variables. Comparisons are made using the Mann–Whitney U and Fisher’s exact tests, as appropriate. Age categories of <55 and >=55 years of age were chosen to focus on clinically relevant early-onset prostate cancer, which is more common in AA men. It is also well-established that Gleason scores <=3 + 4 have a more favorable prognosis since they belong to disease grades 1 and 2, whereas Gleason scores >=4 + 3 have a less favorable prognosis since they belong to disease grades 3, 4, and 5 (as described by the American Cancer Society). Given the size of the cohort, these Gleason score categories were used to derive meaningful clinical correlations with metabolite levels.

#### 2.2.2. Univariate Comparisons of Serum Androgen Metabolite Levels and Clinical Cofactors

Serum androgen metabolite levels were summarized using the mean, standard deviation, and median for each clinical variable. To account for some observations “below the limit of quantification” (BLQ), the mean and standard deviation were estimated using a Tobit model (accounting for left censoring) and the maximum likelihood approach [28]. The association between metabolite levels and each patient characteristic was assessed using a left-censored version of the log-rank test to account for BLQ observations. 

#### 2.2.3. Multivariate Comparisons of Serum Androgen Metabolite Levels

A multivariable Tobit regression model was used to compare metabolites between racial and Gleason groups while adjusting for confounders. Regarding race: age and disease grade were included; for the clinical Gleason grade: disease grade and PSA were included. From the fitted models, least square means estimates of the metabolite levels were obtained with standard errors and compared using a *Z*-test.

#### 2.2.4. Survival Analysis and Hazard Ratios

Time-to-event outcomes (overall and progression-free survival) were summarized using standard Kaplan–Meier (KM) methods, where estimates of 3/5-year rates were obtained with 95% confidence intervals. Overall survival (OS) was defined as the time from radical prostatectomy until death due to any cause or last follow-up. Progression-free survival (PFS) was defined as the time from surgery until persistent disease, disease progression, subsequent treatment, death from prostate cancer, or last follow-up. For patients with persistent disease, the PFS was calculated as 1 day. Persistent disease after radical prostatectomy was defined as follows: 1. PSA levels do not fall to undetectable levels after surgery >0.03–<0.2 ng/mL and is associated with adverse pathological factors (stage T3a or above, diffusely positive surgical margins) and 2. PSA levels >= 0.2 ng/mL. The metabolite levels were dichotomized into low (<=50th percentile) and high (>50th percentile), into terciles, or BLQ and Detectable (i.e., >BLQ). The time-to-event outcomes were then summarized by the categorized metabolite levels using standard Kaplan–Meier methods, with associations evaluated using the log-rank test. The association between survival outcomes and the dichotomized metabolites was examined by race using Cox regression models and fit using Firth’s method. Hazard ratios (HR) for high versus low expression were obtained within the AA and EA cohorts, with corresponding 95% confidence intervals and *p*-values. 

All analyses were conducted in SAS v9.4 (Cary, NC, USA) at a significance level of 0.05.

## 3. Results

### 3.1. Intermediate Metabolites Produced in the Primary Androgen Biosynthesis Pathway Are Associated with Age and Gleason Score in a Race-Independent Manner

We used LC-MS/MS [26] to determine the levels of DHT and T in serum to provide a systemic view of androgen levels. Serum samples were collected under non-fasting conditions. Serum T levels were significantly lower in AA men than in EA men (*p* < 0.05, Figure 1A, Appendix A). Univariate analysis of serum T levels did not show significant correlations with clinical parameters (Appendix A). Additionally, after age and grade adjustments, serum T levels were similar between AA and EA men (Table 1). We did not observe significant differences in serum DHT levels between AA and EA men in univariate and multivariate analysis (Figure 1B, Table 1 and Appendix A). However, serum DHT levels were significantly lower in men with a Gleason score <= 3 + 4 disease compared to a Gleason score >= 4 + 3 in multivariate analysis (Table 1). These data indicate that a low serum DHT level but not T correlates with low Gleason score prostate cancer in a race-independent manner. Serum sex hormone binding globulin (SHBG) binds to DHT/T to reduce the amounts of bioavailable androgens in circulation, therefore, we determined the SHBG level in circulation. No race-specific difference in serum SHBG levels in univariate and multivariate analyses was observed (Table 1 and Appendix A). However, upon adjusting for PSA and grade, the serum SHBG level was lower in men with a Gleason score <=3 + 4 disease compared to a Gleason score >= 4 + 3 (Table 1). Our results show that serum SHBG and DHT levels follow the same trend, i.e., lower in Gleason score <= 3 + 4 disease and higher in Gleason score >= 4 + 3. This suggests that the levels of bioavailable serum DHT are affected by SHBG binding in a similar manner in both Gleason score disease categories.

We next sought to determine the levels of all intermediate metabolites generated during the primary androgen biosynthesis pathway (Appendix A). We first measured serum cholesterol, an essential precursor for Leydig/adrenal and de novo tumor T and DHT synthesis [7]. Serum cholesterol levels may be affected by lipid-lowering agents, but due to the limited sample size, we did not categorize the cohort based on this factor. Serum non-esterified cholesterol levels were similar between AA and EA men and were not significantly associated with clinical variables (Table 1 and Appendix A). This suggests that the above-described differences in DHT levels in men with Gleason scores <= 3 + 4 and >= 4 + 3 are not due to reduced metabolite precursor levels. In the body, cholesterol is converted to pregnenolone [1,2]; in our cohort, 65.8% of AA men (n = 38 AA) vs. 42.0% of EA men (n = 69 EA) had lower serum pregnenolone (*p* < 0.05) (Appendix A). This corresponded with significantly lower serum pregnenolone levels in AA compared to EA men (*p* < 0.01), even after age and grade adjustments (*p* < 0.05) (Figure 1C, Table 1 and Appendix A). One way by which pregnenolone is converted to DHT/T is the primary hydroxypregnenolone-DHEA-ASD androgen biosynthesis pathway [1,2] (Appendix A). Serum hydroxypregnenolone (*p* < 0.01), DHEA (*p* < 0.05), and ASD (*p* < 0.01) levels were significantly higher in men with prostate cancer who are <55 years compared to those >=55 years (Figure 2A, Appendix A). In the multivariate analysis, men with Gleason score >= 4 + 3 disease had significantly higher serum hydroxypregnenolone, DHEA, and ASD levels than men with Gleason score <= 3 + 4 disease (Table 1). These results suggest that higher serum intermediate metabolite levels in the primary androgen biosynthesis pathway are more commonly found in younger men with lower Gleason score disease.

### 3.2. Androgens and Intermediate Metabolite Levels in the Alternate Pathway Are Associated with Clinical Variables of Prostate Cancer

Pregnenolone can be converted to T/DHT through an intermediate progesterone step [1,2] (Appendix A). Serum progesterone (*p* < 0.01, Figure 1D) and associated metabolite levels, including 17α-OH-progesterone (*p* < 0.05, Figure 1E), cortisol (*p* < 0.01, Figure 1F), 20α-dihydro-progesterone (*p* < 0.01, Figure 1G), and corticosterone (*p* < 0.01, Figure 1H), were significantly lower in AA men than EA men (Appendix A). Significant differences in serum 17α-OH-progesterone (*p* < 0.05), cortisol (*p* < 0.001), and corticosterone (*p* < 0.001), in addition to serum deoxycorticosterone (*p* < 0.05) levels between AA and EA men, persisted in the multivariate analysis (Table 1). Aldosterone is one of the terminal metabolites in the progesterone–corticosterone biosynthesis pathway, with CYP11B enzymes regulating the corticosterone conversion to aldosterone [29]. Serum aldosterone levels were above the limit of detection only in a subset of AA men (Table 1 and Appendix A) and were not detected in EA men (Table 1 and Appendix A, *p* < 0.01). Additionally, we found that serum aldosterone levels positively correlated with West African ancestry (Appendix A). Given the positive correlation between West African ancestry and serum aldosterone levels, we investigated if there were differences in CYP11B enzymes in AA and EA men. Exome sequencing analyses of a subset of AA and EA men revealed that AA men with prostate cancer had a significantly higher frequency of multiple SNPs in the genes encoding CYP11B1 and CYP11B2 (Appendix A, Appendix A). Furthermore, two SNPs, rs5313 and rs4544, were in the exonic region of CYP11B2 (Appendix A, Appendix A). Upon analyses of the 1000 Genomes project, we found that in the general population, the exonic CYP11B2 SNPs rs5313 (n = 526 with C > T SNP, total n = 1322) and rs4544 (n = 594 with A > G SNP, total n = 1322) are found in approximately 40–45% of people with African ancestry (data not shown). Compared to the general population, 60% of AA men in the current study present with this SNP, which is significantly higher than the general population (*p* < 0.05, Fischer exact test). We did not find significant differences in cytochrome P450 SNPs, that may affect aldosterone production, between AA and EA men in the current study.

In the progesterone–cortisone pathway, 68.4% and 71.1% of AA men vs. 40.6% and 39.1% of EA men had low serum 17α-OH-progesterone (*p* < 0.01) and progesterone (*p* < 0.01), respectively (Appendix A). Furthermore, in the multivariate analysis with PSA and grade adjustments, significantly higher serum progesterone and related metabolite levels were observed in men with a Gleason score <= 3 + 4 than in men with a Gleason score >= 4 + 3 (Table 1). Serum cortisone levels were significantly higher in men with <=3 + 4 Gleason disease and lower clinical-grade disease (Figure 2B, Appendix A). Serum cortisone levels were also higher in men with no persistent disease than in men with persistent disease (Figure 2C, Appendix A). These results suggest that lower cortisone levels are associated with more aggressive prostate cancer in a race-independent manner. 

DHT synthesis from progesterone occurs via ASD-AND-5α-dione conversions (Appendix A). Serum AND levels were similar between AA and EA men (Table 1 and Appendix A). Significantly, serum AND levels were positively correlated with DHT only in AA men (middle panel Appendix A). Serum AND levels were significantly higher in men with <= 3 + 4 Gleason disease compared with men with >= 4 + 3 Gleason disease when adjusted for PSA and grade (*p* < 0.001) (Table 1). AND can be converted to epiandrosterone, which is further sulfated to serve as a reserve for DHT synthesis [30]. Serum epiandrosterone level was similar between the two racial groups (Table 1 and Appendix A). However, serum epiandrosterone levels were significantly higher in men with >= 4 + 3 Gleason disease compared to men with <= 3 + 4 Gleason disease (Table 1). Since we also observed higher serum DHT levels in Gleason >= 4 + 3 disease, we think that epiandrosterone can also be a source of DHT in men with >= 4 + 3 Gleason disease. These observations demonstrate race- and disease-specific differences in a subset of intermediate androgen metabolites detected in the serum.

### 3.3. Serum Androgen Metabolites Correlate with Progression Free Survival (PFS)

In this study, OS and PFS were similar between AA and EA men (Appendix A). Therefore, we investigated whether serum androgen metabolite levels (high vs. low) can be indicators of disease progression and whether any associations were race specific. The distribution of high and low levels of non-esterified cholesterol was similar between AA and EA men. However, men with high serum non-esterified cholesterol had longer 3-year (*p* < 0.01) and 5-year PFS (*p* < 0.05) compared to men with low serum non-esterified cholesterol (Figure 3A, Appendix A). Therefore, low serum non-esterified cholesterol may be considered a potential race-independent indicator of early disease progression. Lower serum progesterone and a subset of intermediate metabolites were associated with worse PFS. Specifically, the 3-year (*p* = 0.052) and 5-year (*p* < 0.05) PFS were inferior in men with low serum progesterone in comparison with men with high serum progesterone (Figure 3B, Appendix A). The 3-year (*p* < 0.01) and 5-year (*p* < 0.05) PFS were significantly reduced in men with low serum 17α-OH-progesterone in comparison with men with high 17α-OH-progesterone (Figure 3C, Appendix A). Additionally, low serum levels of deoxycorticosterone (*p* < 0.05, Figure 3D) and corticosterone (*p* = 0.094) were associated with reduced 5-year PFS in a race-independent manner (Appendix A). These observations indicate that low progesterone and related metabolite levels, which are more commonly found in AA men, may indicate accelerated disease progression.

In a pooled analysis of AA and EA men, men with low serum ASD, i.e., men in the lower 50 percentile category of dichotomized metabolite levels, had a significantly worse 3- and 5-year PFS than men with high serum ASD (*p* < 0.05) (Figure 3E, Appendix A). Additionally, men with lower serum T levels had significantly reduced 3- and 5-year PFS rates compared to men with higher serum T levels (Figure 3F). Regarding race-specific comparisons, only AA men (left panel) with higher serum T (HR: 0.29(0.10-0.80) vs. 0.95 (0.46–1.99), *p* < 0.05) and pregnenolone levels (HR: 0.21 (0.05–0.84) vs. 1.01 (0.48–2.13), *p* < 0.05) had a significantly decreased risk of PFS compared to EA men (right panel) (Figure 4, Appendix A). These results indicate that serum T and pregnenolone levels are indicators of race-specific PFS of AA and EA men with localized, therapy naïve prostate cancer.

## 4. Discussion

We show that the serum levels of intermediate metabolites are correlated with clinical variables, including the age of disease-onset, Gleason score, and PFS in AA and EA men with localized, therapy naïve prostate cancer. Here, we used the LC-MS/MS, a sensitive and specific technique [26], to identify race-specific differences in the levels of androgens and several intermediate androgen metabolites. LC-MS/MS is based on mass and time of elution and accurately detects the presence of androgen-related metabolites. Previous studies in healthy men have shown that radioimmunoassays detect slightly higher levels of androgens in serum than LC-MS/MS assays [31]. The use of LC-MS/MS in our study could account for the discrepant results from previously published reports that used less sensitive chemiluminescent assays and radioimmunoassays [12,32,33,34]. Therefore, it is vital that androgen and intermediate metabolite measurements in the serum are quantified using reliable and sensitive technologies such as LC-MS/MS. 

Androgen metabolite levels, including DHT and T, can vary depending on whether serum or tissues are analyzed. In our study of localized, therapy-naïve prostate cancer, adjusted serum T and DHT levels did not differ between AA and EA men. These results are similar to the observations of race-specific differences in the levels of intermediate androgen metabolites but not T or DHT levels in prostate cancer tissues in the castration recurrent setting [10,16,17,26]. However, we do find that higher DHT levels are a common occurrence in Gleason score >= 4 + 3 disease in a race-independent manner. In contrast, lower serum levels of androgen intermediate metabolites, cortisol and cortisone, are indicators of Gleason score >= 4 + 3 disease and persistent disease. Serum levels of cortisol and cortisone are lower in AA men with prostate cancer when analyzed using a multivariate approach. Combined, these results suggest that lower serum levels of cortisol and cortisone are indicators of aggressive disease in AA men. Future studies with a larger cohort of AA and EA men with prostate cancer are needed to confirm these race-specific observations. These studies can highlight whether the serum levels of metabolites produced during androgen biosynthesis can be used for prognostic purposes, particularly for AA men with localized, therapy naïve prostate cancer.

It is important to note that the clinical management of prostate cancer, particularly after radical prostatectomy, includes measuring PSA levels in the serum to predict and detect biochemical recurrence [9,35]. In an analogous clinical setting where tissues are not readily available, measuring intermediate androgen metabolites in the serum is more practical and could provide additional insight into disease recurrence than PSA alone. Nevertheless, to fully understand the relationship between serum androgens and androgens in prostatic tissues (healthy and diseased) direct comparisons from the same patients are needed.

Our study is the first that comprehensively characterized the status of androgens in the serum of men with newly diagnosed prostate cancer. Significantly, our results suggest that changes in the androgen biosynthesis pathway are early events in prostate cancer. Previous studies have shown that there are race-specific differences in the gene expression of enzymes regulating androgen biosynthesis as well as SNPs that can potentially alter the function of these enzymes [9,36]. It is possible that the differences in the gene expression of enzymes lead to differences in levels of androgen metabolites in AA and EA men. Serum androgens can be produced by multiple cell types, including the testis and adrenal glands. Whether there are race-specific differences in the gene expression of androgen metabolizing enzymes in the adrenal glands or the testis is unknown. It would be interesting, but technically challenging, to determine whether there are race-specific differences in adrenal glands, the primary source of androgen metabolites in the serum. Our future work will include measurements of serum levels of androgens and intermediate metabolites before disease diagnosis, i.e., men with a higher risk of prostate cancer, and throughout the disease continuum. Additionally, our ongoing work will determine the expression of gene-encoding enzymes that regulate androgen biosynthesis in the same cohort. An ideal study would follow the same cohort of patients and assess race-specific associations between levels of androgen metabolites, including DHT and T, in serum at multiple points during prostate cancer progression. 

In men without cancer, a rapid decline in T levels in the serum is more commonly observed in AA men than in EA men. Significantly, this rapid decline is associated with prostate cancer risk [11,12,37,38,39,40,41]. Whether the targeting of androgen metabolism at earlier stages of prostate cancer would also result in a rapid decline in the DHT/T levels that are predictive of disease progression remains to be investigated.

We discovered a higher frequency of novel germline SNPs in the genes encoding the CYP11B family of enzymes in AA men with prostate cancer. The SNPs we report here occur in introns and exons. SNPs in the exonic region have the potential to affect the coding sequence of CYP11B2 and, ultimately, its function. Other reports have identified specific genomic loci at chromosomes 1 and 13 that are associated with an increased risk of excessive aldosterone production [42]. Excessive aldosterone is usually associated with hypertension. However, there is no evidence that the excessive aldosterone that causes hypertension is linked to prostate cancer risk. Further studies in larger cohorts of different racial groups are needed to establish the causal relationship between the SNPs observed in CYP11B2-encoding genes, corticosterone–aldosterone conversion, and prostate cancer risk.

Interestingly, our pooled analysis of AA and EA men revealed that lower levels of intermediate metabolites, including progesterone, ASD, and T, were associated with PFS. In our study, AA men more often presented with lower levels of these intermediate metabolites. Therefore, our results which show an association of serum intermediate androgen metabolites with PFS are clinically significant. This work also provides a rationale for future measurements of the intermediate metabolites, in addition to DHT and T, in the serum and possibly prostate cancer tissues from a diverse group of men in addition to other racial subgroups. These measurements will determine the utility of intermediate androgen metabolites as indicators of survival outcomes. 

In conclusion, the current study identifies the race- and disease-specific associations of serum androgen metabolites in men with localized therapy naïve prostate cancer. AA men have an increased risk of prostate cancer incidence at a younger age and higher mortality [43,44]. Utilizing serum obtained at the time of diagnosis and determining the levels of androgen metabolites can potentially identify a subset of AA men with inferior prognosis. This may aid in distinguishing men that need increased active surveillance to monitor disease progression. Importantly, our results and other studies [21,45,46,47,48] highlight the importance of including AA men to identify changes and/or molecular markers that are unique to this population of prostate cancer patients.

## Figures and Tables

**Figure 1 metabolites-13-00284-f001:**
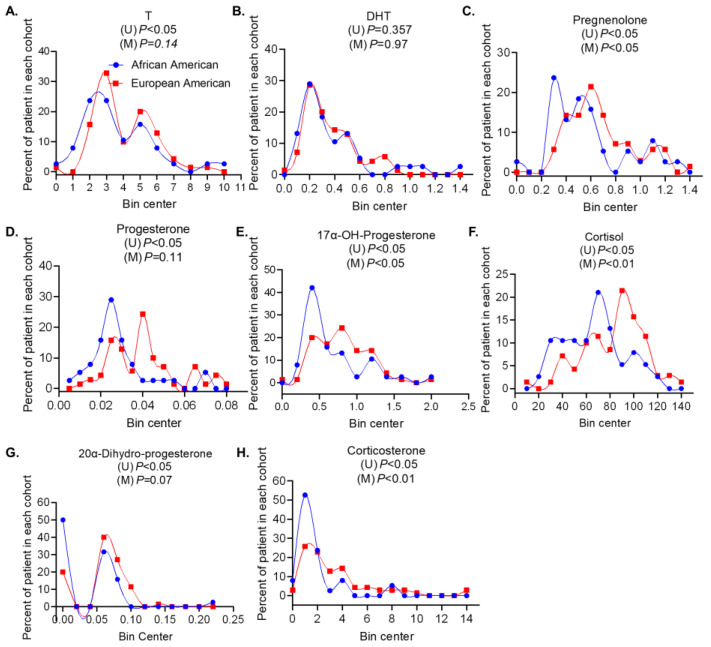
Serum distribution of testosterone (T), dihydrotestosterone (DHT), and intermediate metabolites in AA men and EA men. Differences in the serum metabolite levels between African American (AA) and European American (EA) men were compared using a left-censored version of the log-rank test in the univariate analysis. A multivariable Tobit regression model was used to compare metabolites between racial and Gleason groups while adjusting for confounders. (**A**,**B**) Distribution of serum androgen levels, T and DHT, in AA and EA men with prostate cancer. (**C**–**H**) Distribution of serum levels of intermediate metabolites produced during androgen biosynthesis in AA and EA men with prostate cancer. Blue lines represent the frequency of distribution of T and DHT levels in the serum of AA men and red lines represent the frequency of distribution of T and DHT levels in the serum of EA men. The *p*-value represents significant differences between AA and EA men from the univariate analysis (U) and multivariate analysis (M) that adjusts for age and grade.

**Figure 2 metabolites-13-00284-f002:**
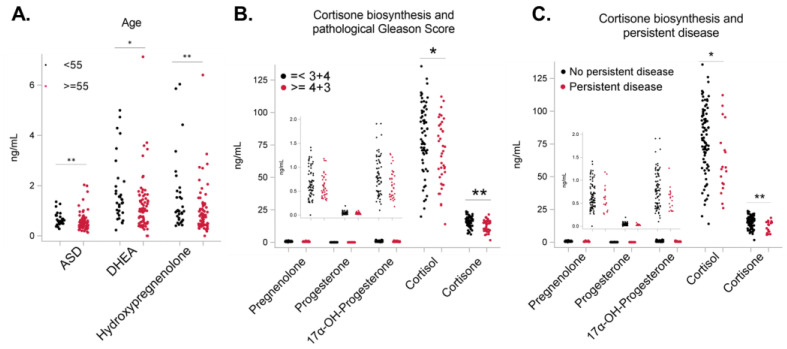
Serum metabolite levels are associated with clinical cofactors. The metabolite levels within groups of clinical cofactors were compared using a left-censored version of the log-rank test. * *p* < 0.05, ** *p* < 0.01 upon comparisons of the mean values in each category. Each dot represents a single clinical serum sample. (**A**) The mean androstenedione (ASD), dehydroepiandrosterone (DHEA), and hydroxypregnonolone serum levels are significantly higher in men <55 years than in men >=55 years with prostate cancer. (**B**,**C**) The mean cortisol and cortisone serum levels are lower in men with a prostate cancer Gleason Score >= 4 + 3 and persistent disease. The insets in each panel have alternate Y-axis to accurately depict the distribution of pregnenolone and 17-α-OH-progesterone.

**Figure 3 metabolites-13-00284-f003:**
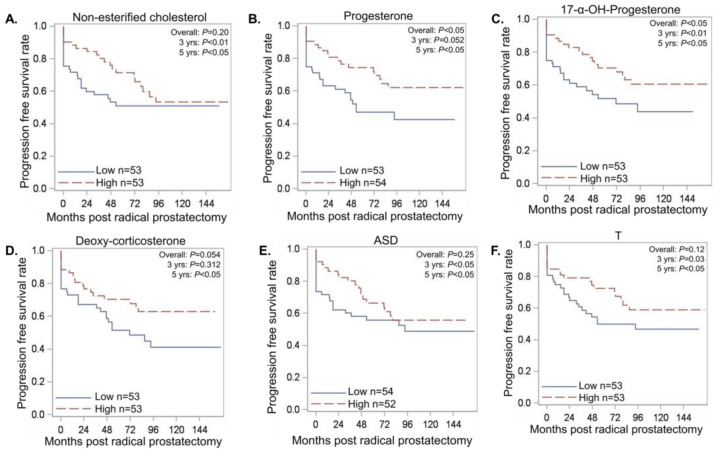
A subset of intermediate metabolites produced during androgen biosynthesis is reflective of a shorter time to disease progression. The metabolite levels were dichotomized into low (<=50th percentile) and high (>50th percentile). The time-to-event outcomes were then summarized by the categorized metabolite levels by using standard Kaplan–Meier methods, and associations were evaluated with standard log-rank tests. (**A**–**F**) Lower levels of intermediate androgen metabolites were found to be associated with a shorter time to disease progression; represented as progression-free survival on the Y-axis. ASD: androstenedione and T: testosterone.

**Figure 4 metabolites-13-00284-f004:**
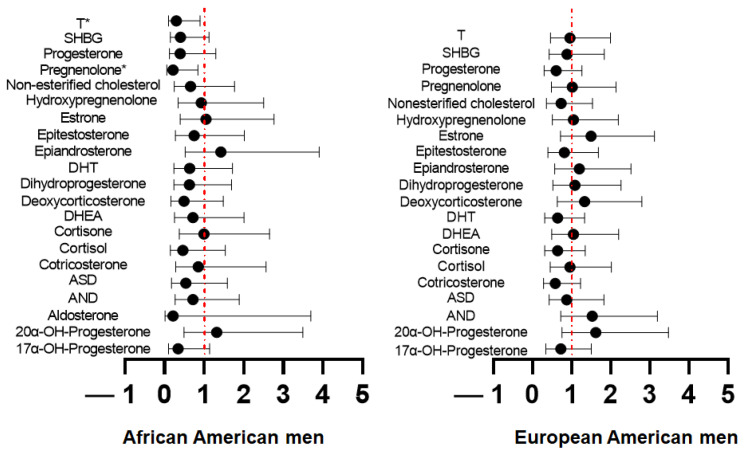
Testosterone (T) and pregnenolone levels are associated with shorter progression-free survival (PFS) only in African American (AA) men. The association between survival outcomes and the dichotomized metabolites was examined by race using Cox regression models and fit using Firth’s method. Higher levels of testosterone and pregnenolone were associated with decreased risk of progression-free survival only in AA men. The circles represent the hazard ratios and the lines represent the 95% confidence intervals. SHBG: Sex hormone binding globulin, DHT: Dihydrotestosterone, DHEA: Dehydroepiandrosterone, ASD: Androstenedione, and AND: All observed Androsterone. * *p* < 0.05.

**Table 1 metabolites-13-00284-t001:** Multivariate analysis of serum androgens and related metabolites of AA and EA men.

Metabolite (Adjusted for Age and Grade)	EA Mean (Std Error)	AA Mean (Std Error)	*p*-Value
17α-OH-Progesterone (ng/mL)	0.83 (0.05)	0.68 (0.06)	0.043
20α-Dihyro-Progesterone (ng/mL)	0.06 (0.00)	0.05 (0.00)	0.071
Aldosterone (ng/mL)	0.13 (0.02)	0.23 (0.03)	0.005
All observed Androsterone (AND) (ng/mL)	0.13 (0.01)	0.12 (0.01)	0.543
Androstenedione (ASD) (ng/mL)	0.64 (0.05)	0.55 (0.05)	0.206
Corticosterone (ng/mL)	3.58 (0.37)	1.86 (0.42)	0.001
Cortisol (ng/mL)	83.18 (3.59)	66.18 (4.03)	0.001
Cortisone (ng/mL)	15.65 (0.63)	14.60 (0.71)	0.254
Dehydroepiandrosterone (DHEA) (ng/mL)	1.62 (0.16)	1.24 (0.18)	0.094
Dihydrotestosterone (DHT) (ng/mL)	0.38 (0.03)	0.38 (0.04)	0.969
Deoxy-Corticosterone (ng/mL)	0.04 (0.00)	0.03 (0.00)	0.032
Dihydro-Progesterone (ng/mL)	0.03 (0.00)	0.03 (0.00)	0.742
Epi-Androsterone (ng/mL)	0.18 (0.02)	0.19 (0.02)	0.836
Epi-Testosterone (ng/mL)	0.04 (0.00)	0.05 (0.00)	0.763
Estrone (ng/mL)	0.03 (0.00)	0.04 (0.00)	0.050
Hydroxy Pregnenolone (ng/mL)	1.48 (0.16)	1.04 (0.18)	0.052
Non-esterified Cholesterol (µg/mL)	97.29 (8.61)	85.21 (9.67)	0.332
Pregnenolone (ng/mL)	0.70 (0.04)	0.59 (0.04)	0.044
Progesterone (ng/mL)	0.04 (0.00)	0.03 (0.00)	0.105
Sex Hormone Binding Globulin (SHBG) (nmol/L)	41.12 (3.03)	42.99 (3.41)	0.670
Testosterone (T) (ng/mL)	4.28 (0.25)	3.73 (0.28)	0.135
**Metabolite (adjusted for PSA and grade)**	**<=3 + 4**	**>=4 + 3**	***p*-value**
17α-OH-Progesterone (ng/mL)	0.82 (0.03)	0.63 (0.07)	0.044
20α-Dihyro-Progesterone (ng/mL)	0.06 (0.00)	0.05 (0.01)	0.025
Aldosterone (ng/mL)	0.17 (0.06)	0.17 (0.13)	1.000
All observed Androsterone (AND) (ng/mL)	0.13 (0.01)	0.07 (0.01)	0.001
Androstenedione (ASD) (ng/mL)	0.66 (0.03)	0.10 (0.06)	<0.001
Corticosterone (ng/mL)	2.82 (0.32)	1.42 (0.65)	0.101
Cortisol (ng/mL)	89.07 (2.69)	41.09 (5.39)	<0.001
Cortisone (ng/mL)	17.64 (0.53)	8.06 (1.07)	<0.001
Dehydroepiandrosterone (DHEA) (ng/mL)	1.59 (0.14)	BLQ (0.28)	<0.001
Dihydrotestosterone (DHT) (ng/mL)	0.39 (0.02)	0.50 (0.04)	0.045
Deoxy-Corticosterone (ng/mL)	0.04 (0.00)	0.02 (0.01)	0.012
Dihydro-Progesterone (ng/mL)	0.03 (0.00)	0.03 (0.00)	0.810
Epi-Androsterone (ng/mL)	0.12 (0.01)	0.19 (0.02)	0.008
Epi-Testosterone (ng/mL)	0.04 (0.00)	0.04 (0.00)	0.885
Estrone (ng/mL)	0.04 (0.00)	0.03 (0.00)	0.284
Hydroxy Pregnenolone (ng/mL)	1.22 (0.14)	BLQ (0.29)	<0.001
Non-esterified Cholesterol (µg/mL)	98.17 (1.44)	96.24 (2.89)	0.614
Pregnenolone (ng/mL)	0.65 (0.03)	0.37 (0.06)	<0.001
Progesterone (ng/mL)	0.04 (0.00)	0.03 (0.01)	0.020
Sex Hormone Binding Globulin (SHBG) (nmol/L)	39.01 (1.65)	55.27 (3.31)	<0.001
Testosterone (T) (ng/mL)	3.97 (0.16)	4.25 (0.31)	0.494

Least square mean estimates of the metabolite levels were obtained with standard errors and compared using a *Z*-test. Regarding race, age and disease grade were included; regarding the Gleason score, disease grade and PSA were included.

## Data Availability

Data will be provided upon reasonable request from the corresponding authors. Data is not publicly available due to privacy or ethical restrictions.

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
