# Peer review of "Serum Androgen Metabolites Correlate with Clinical Variables in African and European American Men with Localized, Therapy Naïve Prostate Cancer"

_metabolites, 2023, doi:10.3390/metabo13020284_

Round 1

Reviewer 1 Report

The present study examines racial disparities in serum androgen concentrations and their relationship with prostate cancer progression. The manuscript is well written, and the figures overall lend support to the main conclusions of the study. As detailed below, my main concerns with the present version of the manuscript relate to the explanation of statistical analyses performed, and in certain cases the presentation of data. With these modifications, the study will likely make an important contribution to the field.

Major points

1. Figure 2, graphs showing serum metabolite levels. The differences between groups are rather small and with overlapping error bars.

a. The first point is that it isn’t clear what the error bars here represent; standard devation? standard error? Second, I could not find detailed information about what statistical tests were performed; multiple Student’s t-tests? Was correction for multiple comparisons applied? This should be detailed in the methods section, and both points should be very briefly mentioned in the figure caption. Without this information, the data is very difficult to interpret. Please revise.

b. Values are aggregated and the averages for each category are represented as bars. Given the relatively small differences mentioned above, would it not be better to show the individual data points together with averaged values shown? This reviewer is concerned that these modest differences may be mostly due to one or a small number of individuals who have exceptionally high or low levels that ‘distort’ the averages shown to yield a presentable p-value, rather than revealing the existence of systematic differences as preferred by the authors. Please revise by including the individual data points together with the bars, perhaps also replacing the bar with more informative boxplots or violin plots.

2. Table 1 (and Supplemental Tables S1-S14). The authors present the results of their multivariate analysis of serum androgens including a column of P-values. However, like discussed above, this reviewer is unable to find how these P-values were derived. Please revise.

3. Supplementary Figure 2 and Discussion. The authors discuss exonic CYP11B SNPs and their “higher frequency” in AA compared with EA men. Please specify what the frequencies of these alleles are based on existing whole genome sequencing data (e.g., 1000 genomes project, or similar). A “very rare” variant could yet be said to have a higher frequency than an “extremely rare” variant, but the key point is whether this difference is likely to provide explanatory power vis-à-vis the authors’ data. If it’s something like 1% vs. 10%, then that would seem plausible. But if it’s something like 0.00001% vs. 0.00002%, then it would seem implausible. Please elaborate.

4. Related to the above point, might cytochrome P450 variants not also (primarily?) affect chemotherapy cancer drug metabolism and/or selection of chemotherapy regimens, thereby confounding the authors' suggestion of an association with corticosterone-aldosterone conversion? Please discuss.

5. Supplemental Table S2. Please specify which ancestry informative markers were used; there are widely different numbers and sets used by different colleagues.

Minor points

6. Figure 3. The authors sub-categorize metabolite levels into low (<50th percentile) and high (>50th percentile). What about the “50th” percentile, was it ignored? Was it counted twice? Or should one of the inequality signs be an “equal to or...” sign? Please clarify.

7. Table and Figure captions throughout. Please re-define acronyms used throughout; figure captions should be interpretable by themselves without needing to search for acronyms in the main body.

Reviewer 2 Report

Swathi Ramakrishnan et al reported the interesting finding of different androgen metabolism in AA and EA man of prostate cancer. They used the high-pressure liquid chromato graphic assay with tandem mass spectral detection and determined serum levels of metabolites produced during DHT/T biosynthesis in African American (AA) and European American (EA) men in localized, therapy naïve prostate cancer. The findings include lower level of serum progesterone and related metabolites in AA men, which were associated with rapid disease progression. The intermediate androgen metabolites between AA and EA men and between men with <=3+4 and >=4+3 23 Gleason score disease were different. They also showed that AA men have higher frequency of single nucleotide polymorphisms in CYP11B1 and CYP11B2, and higher levels of T and pregnenolone were associated with lower risk of progression-free survival. These new findings will be interesting for androgen metabolism and racial disparities and may contribute to the cancer treatment.

The manuscript is overall convincing. The results and the implications will be of interest for cancer researchers and in general to the field. I think the manuscript should be considered for publication, as long as the authors are able to address some specific concerns (see below).

1, It is interesting analysis for the biochemistry synthesis of androgens in AA and EA men, and this manuscript manly focus the process that includes several intermediate emtobolites. Perhaps, it is better to show a figure with these process or pathways and highlight your new findings. 

2, Figure 5, why do you choose 55 year to separate men into two group? Do you consider other age such as 60, 70, 80? Similarly, How about other Gleason score?

3, It is the interesting finding that lower serum levels of cortisol and cortisone are indicators of aggressive disease in AA men. Perhaps, the expression of genes related to the cortisol and cortisone can be analyzed such as in TCGA database et al, which also can verify results of this manuscript.

4, As mentioned in the manuscript “to fully understand the relationship between serum androgens and androgens in prostatic tissues (healthy and diseased) direct comparisons from the same patients are needed.” I think that there were prostate cancer tissues with protein data such as western blot and Immunohistochemistry for many proteins. Therefore, it is better to check these papers and compare their data with this manuscript.

5, This study reported “Serum levels of cortisol and cortisone are lower in AA men with prostate cancer. Combined these results suggest that lower serum levels of cortisol and cortisone are indicators of aggressive disease in AA men.” Could authors discuss the possible reasons of this finding?

6, This study showed interesting finding of the SNPs identified in CYP11B1 and CYP11B2 in AA and EA men. There are many serum androgen and related metabolites, and do you find other interesting SNPs related to other metabolites in AA and EA men, which may be interesting?

Round 2

Reviewer 1 Report

Thank you for your considered revisions. Best wishes for your future research.

Reviewer 2 Report

The revised manuscript solved my questions, and I think that it should be accepted.